# Simple and Efficient Pressure Ulcer Reconstruction via Primary Closure Combined with Closed-Incision Negative Pressure Wound Therapy (CiNPWT)—Experience of a Single Surgeon

**DOI:** 10.3390/jpm12020182

**Published:** 2022-01-29

**Authors:** Kuo-Feng Hsu, Li-Ting Kao, Pei-Yi Chu, Chun-Yu Chen, Yu-Yu Chou, Dun-Wei Huang, Ting-Hsuan Liu, Sheng-Lin Tsai, Chien-Wei Wu, Chih-Chun Hou, Chih-Hsin Wang, Niann-Tzyy Dai, Shyi-Gen Chen, Yuan-Sheng Tzeng

**Affiliations:** 1Division of Plastic and Reconstructive Surgery, Department of Surgery, Tri-Service General Hospital, National Defense Medical Center, Taipei 114202, Taiwan; captain0416@gmail.com (K.-F.H.); illusioncjy@gmail.com (C.-Y.C.); justyu1112@gmail.com (Y.-Y.C.); gdream12731@gmail.com (D.-W.H.); liutingxuan@gmail.com (T.-H.L.); woodytsai2012@gmail.com (S.-L.T.); alex199180220@hotmail.com (C.-W.W.); i0937361709@hotmail.com (C.-C.H.); super-derrick@yahoo.com.tw (C.-H.W.); niantzyydai@gmail.com (N.-T.D.); shyigen@gmail.com (S.-G.C.); 2Department of Surgery, Tri-Service General Hospital, National Defense Medical Center, Taipei 114202, Taiwan; chu.peiyi.88@gmail.com; 3Department of Pharmacy Practice, Tri-Service General Hospital, National Defense Medical Center, Taipei 114202, Taiwan; kaoliting@gmail.com; 4Graduate Institute of Life Sciences, National Defense Medical Center, Taipei 114202, Taiwan

**Keywords:** pressure ulcer, surgical reconstruction, dressing materials, wound control, negative pressure wound therapy, vacuum-assisted closure, closed-incision negative pressure wound therapy, closed-incision negative pressure therapy

## Abstract

Background: In this study, we aimed to analyze the clinical efficacy of closed-incision negative pressure wound therapy (CiNPWT) when combined with primary closure (PC) in a patient with pressure ulcers, based on one single surgeon’s experience at our medical center. Methods: We retrospectively reviewed the data of patients with stage III or IV pressure ulcers who underwent reconstruction surgery. Patient characteristics, including age, sex, cause and location of defect, comorbidities, lesion size, wound reconstruction methods, operation time, debridement times, application of CiNPWT to reconstructed wounds, duration of hospital stay, and wound complications were analyzed. Results: Operation time (38.16 ± 14.02 vs. 84.73 ± 48.55 min) and duration of hospitalization (36.78 ± 26.92 vs. 56.70 ± 58.43 days) were shorter in the PC + CiNPWT group than in the traditional group. The frequency of debridement (2.13 ± 0.98 vs. 2.76 ± 2.20 times) was also lower in the PC + CiNPWT group than in the traditional group. The average reconstructed wound size did not significantly differ between the groups (63.47 ± 42.70 vs. 62.85 ± 49.94 cm^2^), and there were no significant differences in wound healing (81.25% vs. 75.38%), minor complications (18.75% vs. 21.54%), major complications (0% vs. 3.85%), or mortality (6.25% vs. 10.00%) between the groups. Conclusions: Our findings indicate that PC combined with CiNPWT represents an alternative reconstruction option for patients with pressure ulcers, especially in those for whom prolonged anesthesia is unsuitable.

## 1. Introduction

Pressure ulcers are associated with specific positions that place prolonged pressure on the skin and underlying soft tissue (e.g., sacral ulcers in the supine position, ischial ulcers in the sitting position), occurring most frequently in patients with spinal cord injury, dementia, or other conditions involving long-term immobilization. Currently, treatment for pressure ulcers remains challenging, and their complex etiology necessitates a multidisciplinary approach. Since pressure ulcers are generally large, direct excision and primary closure (PC) of pressure ulcers can lead to wound dehiscence due to tension across the wound edge. Therefore, a large amount of vascularized tissue is required to completely fill the dead spaces that remain following the removal of extensive ulcers. Although the value of serial surgical debridement for eradicating infection following flap reconstruction has been well established [1,2,3,4], the time required for repeated surgical treatment and flap reconstruction increases the anesthetic risk, which may increase perioperative complication risk.

Negative pressure wound therapy (NPWT) provides controlled continuous or intermittent sub-atmospheric pressure over an open wound, and has recently been widely used for temporization of traumatic wounds, as well as treatment of difficult wounds, such as pressure ulcers [5,6,7]. Previous studies have demonstrated that application of NPWT to a wound sutured under tension may decrease tension across the suture site by distributing the traction forces concentrated at the suture point to the skin flap, and by increasing skin flap perfusion [8,9,10,11,12,13,14,15,16,17]. In this manner, NPWT effectively approximates both skin flaps, increases tissue perfusion, reduces the dead space in the wound, and promotes marginal apposition of the wound edge, thereby improving the success rate of wound healing in patients undergoing direct closure.

Numerous studies have reported the application of closed-incision NPWT (CiNPWT) in patients with various types of wounds, including those resulting from sternotomy, laparotomy, vascular, orthopedic, and reconstructive surgeries [18,19,20,21,22,23,24,25,26,27]. However, to our knowledge, no studies have reported its application in patients undergoing pressure ulcer reconstruction. Given the dermatotraction effect of CiNPWT, it may allow for PC during pressure ulcer reconstruction in patients not deemed fit for prolonged general anesthesia.

In this study, we retrospectively analyzed the clinical efficacy of PC + CiNPWT as an alternative to flap surgery in patients with pressure ulcers, based on experiences of a single surgeon at our center.

## 2. Methods

### 2.1. Patients and Device Application

From October 2019, PC + CiNPWT for pressure ulcer reconstruction was performed at Tri-service General Hospital (Taipei City, Taiwan) by a single surgeon in the Division of Plastic and Reconstructive Surgery, Department of Surgery. Patient eligibility criteria for this procedure included no residual necrotic tissue after wound debridement, good infection control, good nutritional support, and alignment of the wound edge after manual pinching or pushing, even if blanching response had been lost at the wound edge due to tension.

Antibiotics were administered until suture removal in patients with infected pressure ulcers. Patients also underwent consultation with an infection specialist, who adjusted the antibiotic and dosage based on the result of bacterial cultures and the clinical response.

When necessary, multiple surgical debridement procedures were performed under anesthesia to reach the aforementioned reconstruction indication. The interval between each debridement was 3–7 days, which was adjusted according to the patient’s clinical condition and operating room availability. Before surgical reconstruction of the wound, the open wound could be temporized with NPWT as a bridge to reconstruction; otherwise, wound dressing was performed with diluted betadine wet dressing. The choice of PC + CiNPWT or traditional flap surgery for pressure ulcer reconstruction was determined by the attending physician, based on an evaluation of the wound before surgery. Anesthesia type was determined by a specialist anesthesiologist, based on the patient’s physical condition, surgical site, posture, and surgery. General and spinal anesthesia both met our surgical needs, although the operation could complete under light sedation combined with tumescent anesthesia in cooperative patients. The wound underwent a final debridement prior to closure, which involves cutting off part of the wound edge in a fusiform manner using a sharp scalpel, creating a fresh wound edge to facilitate subsequent suturing and healing. A closed-suction drain was placed in the wound to reduce seroma formation. Surgical closure of the wound was performed using a standard “layer by layer” strategy. The CiNPWT system was applied after wound closure (Prevena^®^ or Vacuum Assisted Closure^®^ (V.A.C.^®^), KCI USA, Inc., San Antonio, TX, USA), with a pressure level setting of −125 mmHg according to its design. When using VAC, the polyethylene foam was cut to a size approximately 3 cm larger than the sutured wound edge on both sides, and a hydrofiber dressing (Aquacel^®^, ConvaTec, Princeton, NJ, USA) was placed between the wound and the foam to reduce wound skin maceration. In contrast, per its design, Prevena was directly applied to the wound. Depending on the amount of discharge from the wound, the CiNPWT unit was disassembled 7–14 days after installation. The decision to continue CiNPWT use was determined based on the status of wound healing.

All patients received the same postoperative care, which involved keeping the wound clean, turning over regularly to avoid continuous pressure on the wound, maintaining proper nutrition, and control of underlying medical problems. The closed-suction drain was removed 7–14 days after operation once the drainage fluid had become clear, and did not exceed 30 mL/day. Patients were discharged upon meeting the following criteria: beginning of wound healing (checked after disassembly of the CiNPWT), clearing of the drainage tube and low drainage volume, good nutritional status, no requirement for parenteral nutrition support, improvement and control of infection using oral antibiotic, control or treatment of underlying medical issues, understanding of the care required by the patient’s family. The drainage tube and second CiNPWT (when present) were removed in the outpatient clinic. Follow-up visits were performed at the outpatient clinic 1–2 weeks after discharge from the hospital, and 2–3 times in the subsequent months.

### 2.2. Statistical Analysis

To investigate the clinical benefits of PC + CiNPWT, we retrospectively reviewed data for patients with stage III or IV pressure ulcers who underwent surgery performed by a single surgeon at Tri-Service General Hospital, Taiwan, between January 2011 (when our hospital began to implement electronic operation reports, allowing us to obtain photos of surgery) and March 2021. All patients were admitted by or had consulted this plastic surgeon, who specialized in pressure ulcers. A total of 157 patients who underwent pressure ulcer reconstruction were initially identified from the electronic report system. We reviewed photographs obtained during surgeries as well as medical records, and 29 patients were excluded due to incomplete records (missing wound reconstruction photographs, incomplete or missing medical records of the wound healing process during admission or at the outpatient department, and missing photos of wound complications when applicable). Overall, 128 patients with 162 pressure ulcers were included. Patient characteristics, including age, sex, cause and location of defects, comorbidities, wound cultures, lesion size, wound reconstruction methods, CiNPWT application, duration of hospital stay, and wound complications were assessed (Appendix A). Of the 128 patients, 68 were female and 60 were male, with an age range of 20–97 years (Table 1).

Wounds were categorized into seven groups according to reconstruction methods (Table 2): gluteus maximus myocutaneous V-Y advancement flap (VY), superior gluteal artery perforator flap (SGAP), pedicle anterolateral thigh flap (pALT), tensor fascia lata flap (TFL), PC, other, and PC + CiNPWT. All statistical analyses were performed using SPSS 20 software (IBM Corp., Armonk, NY, USA). Chi-square tests or Fisher’s exact tests were used to compare the distributions of sex, comorbidities, debridement times, defect locations, and wound outcomes between groups. One-way analyses of variance (ANOVA) or independent *t*-tests were performed to estimate differences in patient age, wound size, operation time, and hospital stay between groups. A two-tailed *p*-value of <0.05 indicated statistical significance.

### 2.3. Clinical Outcomes

To evaluate clinical outcomes, we defined “healed” as adequate wound healing with suture removal 14–21 days after closure and no occurrence of dehiscence within the following 3 months. Minor complications were defined as poor wound healing or partial dehiscence requiring a second debridement and PC, whereas major complications required another flap reconstruction.

### 2.4. Representative Cases

#### 2.4.1. Case 1

An 84-year-old woman, bedridden due to cerebral infarction, developed a grade IV sacral pressure ulcer. After initial debridement and NPWT application to prepare the wound bed, the wound was directly excised, resulting in a 10 × 13 cm (width × length) defect. The wound was reconstructed directly via PC + CiNPWT (Prevena). Both surgical procedures were performed in the lateral decubitus position. Seven days after reconstruction, the CiNPWT unit was removed, and the wound healed well with no skin maceration over the wound edge. The nylon suture was removed 2 weeks later, and we observed that the wound had recovered without dehiscence or recurrence 1.5 months after reconstruction (Figure 1).

#### 2.4.2. Case 2

A 52-year-old man, bedridden due to a ruptured cerebral aneurysm, gradually developed two communicating pressure ulcers over the bilateral posterior superior iliac spine region. After initial debridement, reconstruction surgery was performed under general anesthesia in the prone position. We performed a fusiform excision of unviable tissue, resulting in a 6 × 17 cm fresh wound, and reconstruction was achieved via PC + CiNPWT (Prevena). The wound was closed with a superficial fascia suture and a subdermal suture (2-0 and 3-0 VICRYL^®^ (polyglactin 910)), and the epidermal nylon suture was omitted; subsequently, Prevena was applied directly. The bilateral wound edges were well approximated, and no maceration was observed when the Prevena was removed 7 days postoperatively. The wound had healed well by the 2-month follow-up (Figure 2).

#### 2.4.3. Case 3

A 69-year-old woman, bedridden due to underlying hydrocephalus, was admitted for trochanteric pressure ulcers. After initial debridement, the wound was an 8 × 7 cm grade IV lesion with a depth up to the tensor fascia lata. We excised the unviable tissue and reconstructed the wound with a primary suture, following which CiNPWT (VAC) was applied. We also applied Aquacel Ag^+^ extra (ConvaTec, Princeton, NJ, USA) hydrofiber dressings (light-gray dressing material, Figure 3d) between the wound and polyethylene foam to prevent skin maceration. Examination on postoperative day 7 revealed that the wound was well approximated, without maceration and with a significant reduction in tissue edema. At the 1-month follow-up, we observed that the wound had healed well, and there was no dehiscence or recurrence.

#### 2.4.4. Case 4

A 95-year-old woman, bedridden due to senile dementia, was admitted for a pressure ulcer over the bilateral trochanter and sacral region (a: sacral and left trochanter, c: right trochanter). Pressure ulcers over the left trochanteric region were reconstructed using a pedicle ALT flap, while those over the sacral and right trochanteric regions were reconstructed via fusiform excision and closed primarily with CiNPWT (VAC). All three wounds healed well. The wound on the right trochanteric region was closed under tension when compared with that in the left. Eventually, the edematous soft tissue subsided, and the tissue tension decreased due to the dermatotraction effect of CiNPWT. Obviously, as the PC + CiNPWT procedure was less difficult, the length of the wound was significantly shorter than that for the pALT flap (Figure 4).

## 3. Results

### 3.1. Patient Characteristics

A total of 128 patients, including 60 men (46.88%) and 68 women (53.13%), with an age range of 20–97 years (mean age, 75.80 ± 14.71 years), were included in the analysis (Table 1). The most common etiology was dementia (*n* = 36, 28.13%), followed by cerebrovascular accident (*n* = 30, 23.44%) and Parkinson’s disease (*n* = 14, 10.94%). A total of 162 wounds were analyzed (*n*_T_ = 162), with most defects located in the sacral region (*n*_T_ = 114, 70.37%), followed by the trochanteric (*n*_T_ = 31, 19.14%) and ischial regions (*n*_T_ = 7, 4.32%).

### 3.2. Comparison of Overall Outcomes

Statistical analyses revealed no significant differences in sex, age, or comorbidities between groups. The average follow-up time among all patients was 17.95 ± 16.92 months. The mean follow-up time in each group is shown in Table 2. Patients were grouped based on the first wound treated (Table 2, upper, *n* = 128). Comorbidities were counted based on the number of patients rather than the number of ulcers, to avoid overcounting. However, when patients were grouped based on the second wound treated, there were still no significant differences in comorbidities between groups.

When grouped according to wounds (Table 2, lower, *n*_T_ = 162), the reconstructed wound size, operation time, and debridement times different significantly between the groups (*p* < 0.05). Reconstructed wound size was largest in the SGAP group (103.29 ± 36.31 cm^2^) and smallest in the PC group (35.03 ± 23.13 cm^2^). Reconstruction time was longest in the pALT group (174.88 ± 45.16 min) and shortest in the PC + CiNPWT group (38.16 ± 14.02 min). The pALT group underwent the most debridement (4.50 ± 2.39 times), whereas the PC + CiNPWT group underwent the least (2.13 ± 0.98 times).

Additionally, the selected flap type was strongly correlated with the reconstruction site (Table 2). The VY flap was selected for sacral and trochanteric areas; the SGAP flap was selected only for sacral sores; the pALT was selected for the trochanteric and ischial regions; and the TFL flap was selected only for the trochanteric region. PC and PC + CiNPWT were selected for all regions, although most wounds were in the sacral and trochanter regions.

### 3.3. Comparison of Outcomes between the Traditional and PC + CiNPWT Groups

Table 3 shows that there were no significant differences in average reconstructed wound size between the traditional reconstruction and PC + CiNPWT groups (63.47 ± 42.70 vs. 62.85 ± 49.94 cm^2^, *p* = 0.9490), although the PC + CiNPWT group had a shorter operation time (38.16 ± 14.02 vs. 84.73 ± 48.55 min, *p* < 0.001), underwent fewer debridements (2.13 ± 0.98 vs. 2.76 ± 2.20 times, *p* = 0.0153), and had a shorter duration of hospitalization (36.78 ± 26.92 vs. 56.70 ± 58.43 days, *p* = 0.0054) than the traditional group. There were no significant differences in sex, age, or outcomes between groups.

There were no significant differences in the frequency of healed wounds (81.25% vs. 75.38%, *p* = 0.4830), minor complications (18.75% vs. 21.54%, *p* = 0.7286), or mortality between the groups (6.25 vs. 10.00%, *p* = 0.7374).

There were also no significant differences in the sites of reconstruction between groups, with most occurring in the sacral region, followed by the trochanteric, ischial, and back regions.

### 3.4. Comparison of Outcomes between the PC and PC + CiNPWT Groups

There were no significant differences in operation time, number of debridements, hospital days, or outcomes between the PC and PC + CiNPWT groups (Table 4). The reconstructed wound size was significantly larger in the PC + CiNPWT group than in the PC group (63.47 ± 42.70 cm^2^ vs. 35.03 ± 23.13 cm^2^, *p* = 0.0019).

There were significant differences in data related to reconstructed sites. Statistical results revealed that the top three reconstruction sites in the two groups were the sacral, trochanteric, and ischial regions, yet the proportions differed between the groups. Notably, the numbers of patients requiring a PC + CiNPWT incision were 1, 1, and 0 among those with ulcers in the ischial, back, and other regions, respectively. The small number of cases may have affected the results, highlighting the need for further research.

## 4. Discussion

Our results indicated that PC + CiNPWT was associated with fewer debridement procedures, shorter operation times, and a shorter duration of hospitalization than other traditional forms of pressure ulcer reconstruction surgery. Furthermore, the sizes of the reconstructed wounds were similar between the PC + CiNPWT and traditional reconstruction groups, and there were no significant differences in wound prognosis between the groups. As PC + CiNPWT is a simpler procedure that shortens the operation time, it may be useful for pressure ulcer reconstruction given its ability to prevent prolonged general anesthesia.

Currently, treatment for pressure ulcers remains challenging, and their complex etiology requires a multidisciplinary approach [4,28,29]. Conservative management for pressure ulcers is ineffective for deep or large defects (stage III and IV pressure ulcers), and effective treatment requires infection eradication via aggressive debridement and surgical wound reconstruction. Standard methods for daily dressings and serial debridement require prolonged hospitalization. Wound-coverage procedures using a surgical flap after pressure relief and adequate nutrition are indicated in patients with pressure ulcers resistant to conservative management. Various flaps, such as the gluteus maximus myocutaneous flap designed in a V-Y pattern or rotation fashion, exist for sacral-pressure ulcer reconstruction [1,3,30,31]. Recently, the perforator flap has played an important role in reconstructions because of lower donor-site morbidity and free rotational arc [2,32,33,34,35]. For a single flap, the reported maximum reconstruction area for SGAP flaps can reach 12 × 14 cm [2], whereas it can reach 12 × 12 cm for bilateral gluteus maximus myocutaneous V-Y advancement flaps [1]. However, operation times for flap reconstruction surgery are lengthy [2,33,36], and require general anesthesia. Moreover, for the most common pressure ulcers occurring in the sacral region, flap reconstruction surgery should be performed in the prone position with intubation. This is undoubtedly challenging in patients who are old, have multiple comorbidities, or are in poor general condition, which is the case in most patients with pressure ulcers. Smaller pressure ulcers can be simply excised and primarily sutured under local anesthesia; however, they may be prone to dehiscence due to factors such as skin tension. Therefore, this option is not preferred for the reconstruction of larger pressure ulcers.

Traditionally, if the wound is sutured directly, tight skin tension may occur, and flap reconstruction surgery will be subsequently chosen. With the introduction of CiNPWT, wounds previously considered problematic due to excessive skin tension can be confidently sutured without concerns related to wound dehiscence. Even if subsequent wounds exhibit partial dehiscence, the tight wound edges will eventually relax due to the dermatotraction effect of NPWT; accordingly, most dehiscence wounds can be reconstructed via direct wound excision following a second PC + CiNPWT procedure. We previously published a retrospective report of 117 stage III/IV pressure ulcers treated between 2010 and March 2019 [37], before CiNPWT was used on pressure ulcer reconstruction. In that study, the mean defect area of PC was 16cm^2^. A total of 34 wounds (34/117, 24.6%) were closed with PC, with two cases of partial dehiscence healed by secondary healing (5.9%) and five cases of total dehiscence that required a secondary operation (14.7%). PC may be considered if the defective area is <16 cm^2^. In the current study, the patient population partially overlapped with that of previous study (except PC + CiNPWT cases, who were all treated after October 2019). The mean reconstructed area in the PC group was 35.03 ± 23.13 cm^2^, larger than the 16 cm^2^. This may have been due to the small sample size (PC, *n*_T_ = 30), and the differences in wound size of several samples may cause statistical discrepancies. Alternatively, PC + CiNPWT may have increased the surgeon’s confidence in wound closure, indirectly affecting wound size even in the absence of CiNPWT. However, current evidence suggests that CiNPWT can reduce surgical site infection, although there is limited evidence to support that CiNPWT can reduce dehiscence [24,38,39,40,41]. Further research is required to verify our findings regarding the latter.

Furthermore, although there was no significant difference in the average size of the reconstructed wound between PC + CiNPWT and other reconstruction methods, this likely resulted from the relatively small wound sizes in the PC, TFL, VY, and other groups. The SGAP and pALT groups had the largest reconstructed wound sizes, longest operation times, and highest number of debridement procedures. Comparison among the SGAP, pALT, and PC + CiNPWT groups alone may lead to different results. In terms of average reconstruction area, PC + CiNPWT may be more suitable for replacing VY, TFL, and other surgical flaps with similar reconstruction areas for certain, well-chosen wounds.

The potential perioperative complications of general anesthesia, including cardiopulmonary, renal, and neurologic (postoperative cognitive dysfunction) complications, have been well studied, and more likely affect older patients and patients with multiple comorbidities [42,43,44,45,46,47], most of whom are prone to developing pressure ulcers. Induction of general anesthesia can also be challenging in older adults, given that aging is associated with several mental and physiological differences, as well as frequent coexistence of diseases. Therefore, many studies have suggested that spinal anesthesia can be used in lower limb surgery (e.g., hip, knee, and revascularization surgery) instead of general anesthesia [48,49,50,51,52]. Moreover, some studies have reported that spinal anesthesia results in lower rates of surgical complications, re-operations, and intensive care unit admissions, and may decrease 30-day mortality and the duration of hospitalization in patients undergoing lower limb surgery [48,50].

The direct suturing of wounds simplifies wound reconstruction and reduces operation times. In contrast, flap operations require approximately 1 h for flap elevation, and the relatively greater wound length requires a longer time for wound closure, resulting in a total procedure time of >2 h in most cases [2,33]. According to our results, the average operation time for PC was 38 min, which is markedly shorter than the 84 min for overall traditional reconstruction (Table 3). This can not only shorten ventilator use duration during general anesthesia, but also allow for the patient to undergo reconstructive surgery under spinal anesthesia due to the short operation time, further reducing anesthetic risk.

Notably, it is possible that differences in debridement times and hospitalization length may have been due to differences in general condition and comorbidities among patients. Patients whose wounds can be reconstructed with PC may be in a relatively better condition, with less severe wounds and fewer comorbidities. However, our statistical analyses revealed no differences in sex, age, or comorbidities between groups. Therefore, on considering the objective data, it is evident that general condition, wound condition, and comorbidities are not the main factors affecting debridement times and hospitalization length. In a typical clinical setting, NPWT is used for temporalized wound coverage between debridements when affordable for the patient. Many studies have demonstrated that NPWT can shorten wound healing time [26], indicating that an associated decrease in the length of hospitalization may have biased our results. However, we did not collect data regarding the use of NPWT or its duration, necessitating a focus on these variables in future studies.

Osteomyelitis is another clinical condition that may influence debridement time, reconstruction selection, and the duration of hospitalization. Most cases of osteomyelitis related to pressure ulcers are caused by bone exposure and wound infection. In our experience, osteomyelitis mostly exists in the surface cortical bone, although more severe cases may involve invasion to the medullary bone. However, during debridement operations before reconstruction, we attempt to remove these sequestrum to create a viable wound bed. Most patients undergoing this procedure exhibit good results when provided with appropriate systemic antibiotic treatments. In our center, routine treatment for infection in patients with pressure ulcers includes administration of antibiotic treatment along with several debridement operations (approximately 2–3 times, although more procedures may be required in difficult cases), and antibiotic administration is continued until suture removal. Generally, patients treated with this protocol receive antibiotic treatment for more than 4–6 weeks, with an approximately 1-week interval between each debridement and 2–3 weeks from reconstruction to suture removal. Our treatment method is also consistent with those described in previous studies [53,54]. In this study, we did not routinely examine whether patients had osteomyelitis. Therefore, the presence of combined osteomyelitis in the wound was not a factor when choosing PC + CiNPWT for wound reconstruction. However, when severe osteomyelitis is present, most wounds are large, with large areas of bone exposure, meaning that simple reconstruction via PC + CiNPWT may not be possible. In this study, we focused on the ability of PC + CiNPWT to shorten the operation time and reduce the risk of anesthesia. Additional studies are required to determine whether osteomyelitis affects prognosis in patients undergoing various types of pressure ulcer reconstruction.

Based on the abovementioned information, we speculated that PC + CiNPWT represents an alternative to traditional flap reconstruction for patients in whom prolonged general anesthesia is unsuitable, given its advantages in terms of debridement time, operation time, and the duration of hospitalization.

Other factors that may affect reconstruction include the quality of care provided by caregivers, family support, and family socioeconomic status. Better quality of care may be related to better wound care and the frequency of turning over, which can directly affect the prognosis of wound reconstruction. Moreover, both family support and socioeconomic status directly and indirectly affect quality of care. However, there were no relevant data on the abovementioned factors in this study, highlighting the need for future studies to address these issues.

The present study had some limitations, including its retrospective observational nature and reliance on a single surgeon’s experience. A surgeon’s skills will improve over time, potentially influencing statistical results. The present study utilized a long retrospective interval because numbers of cases involving SGAP, pALT and TFL flaps were relatively small, requiring more time to accumulate cases. In addition, the recurrence rate represents an important issue in pressure ulcer reconstruction, and patients treated using myocutaneous flaps are believed to be more resistant to pressure ulcer recurrence. In our study, only six PC + CiNPWT wounds were treated for more than 1 year, making it difficult to reach a definitive conclusion regarding recurrence following PC + CiNPWT. As PC + CiNPWT was also performed only in selected cases, further studies involving longer follow-up periods and prospective randomization are required.

## 5. Conclusions

Our findings indicated that direct excision of pressure ulcers and reconstruction via PC with simultaneous CiNPWT may be an alternative to traditional reconstruction surgery, as this method reduces the difficulty of the operation, operation times, and anesthetic risk. However, there is insufficient evidence supporting NPWT use for tensile wound closure, and our conclusion is based on a retrospective observational analysis. Although further studies are required, our findings suggest that PC + CiNPWT may be useful for pressure ulcer reconstruction in patients deemed unsuitable for surgery involving prolonged anesthesia.

## Figures and Tables

**Figure 1 jpm-12-00182-f001:**
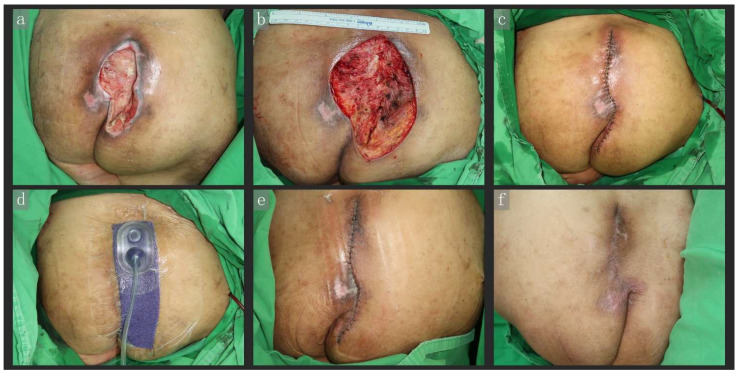
An 84-year-old woman, bedridden due to cerebral infarction, developed a grade IV sacral pressure ulcer. After the first debridement (**a**), the wound was directly excised, resulting in a 10 × 13 cm (width × length) defect (**b**). The wound was reconstructed with primary closure (**c**) and CiNPWT (Prevena) (**d**). Reconstruction surgery was performed with the patient in the left lateral decubitus position. From the photo, it can be seen that there is a little tension across the sutured wound. After 7 days, the CiNPWT unit was removed, and the wound had healed well, with no skin maceration over the wound edge (**e**). The wound had healed without dehiscence or recurrence by 1.5 months after reconstruction (**f**). Abbreviations: CiNPWT, closed-incision negative wound pressure therapy.

**Figure 2 jpm-12-00182-f002:**
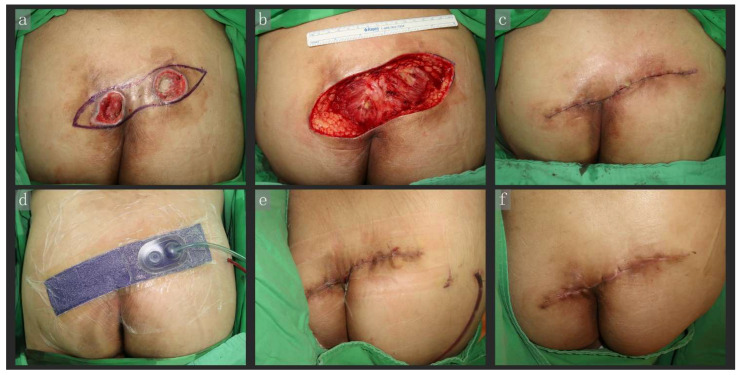
A 52-year-old man, bedridden due to a ruptured cerebral aneurysm, gradually developed two communicating pressure ulcers over the bilateral posterior superior iliac spine region. After debridement, we performed a fusiform excision of unviable tissue (**a**), resulting in a 6 × 17 cm fresh wound (**b**), and reconstruction was achieved via primary closure and CiNPWT (Prevena) (**c**,**d**). The wound was closed only with a superficial fascia suture and subdermal suture, while the epidermal nylon suture was omitted (**c**). Subsequently, the Prevena system was applied directly. As shown in the image, the edges of the bilateral wounds were well approximated, and no maceration was observed when the Prevena system was removed 7 days postoperatively (**e**). The wound had healed well by the 2-month follow-up at (**f**). Abbreviations: CiNPWT, closed-incision negative wound pressure therapy.

**Figure 3 jpm-12-00182-f003:**
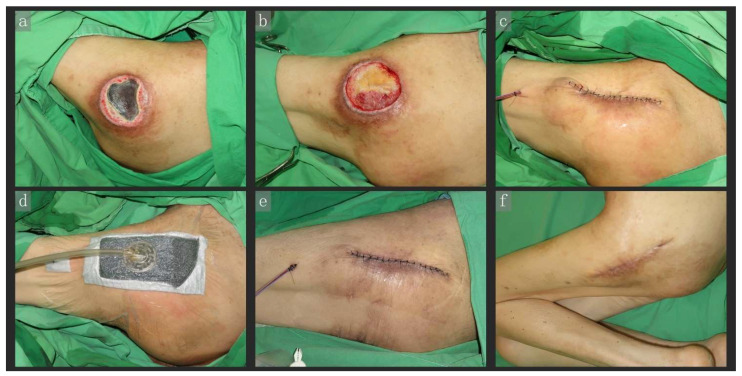
A 69-year-old woman, bedridden due to underlying hydrocephalus, was admitted to our hospital for a trochanteric pressure ulcer (**a**). After initial debridement, the wound presented as an 8 × 7 cm grade IV lesion with a depth up to the tensor fascia lata (**b**). We excised the unviable tissue and performed primary wound closure (**c**), subsequently covering it with the CiNPWT unit (VAC). The light-gray dressing material between the wound and polyethylene foam was Aquacel Ag^+^ extra hydrofiber dressing (**d**). Examination of the wound on postoperative day 7 indicated that it was well approximated, without maceration. Additionally, tissue edema was significantly reduced postoperatively when compared with that in photo C (**e**). At the 1-month follow-up, the wound had healed well, with no dehiscence or pressure ulcer recurrence (**f**). Abbreviations: CiNPWT, closed-incision negative wound pressure therapy; VAC, vacuum-assisted closure.

**Figure 4 jpm-12-00182-f004:**
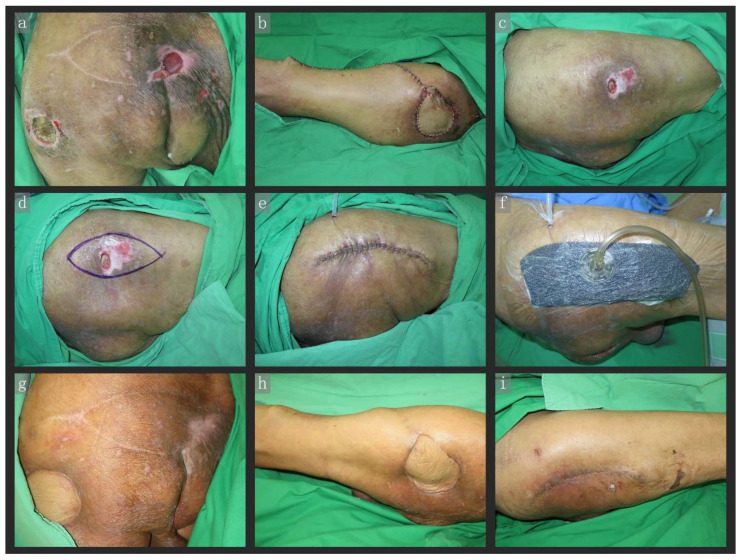
A 95-year-old woman, bedridden due to senile dementia, was admitted for pressure ulcers over the bilateral trochanter and sacral region (**a**: sacral and left trochanter, **c**: right trochanter). The pressure ulcer over the left trochanteric region was reconstructed using a pALT flap (**b**), while those over the sacral and right trochanteric regions were reconstructed via fusiform excision and PC + CiNPWT (VAC) (**c**–**f**). All three wounds healed well (**g**–**i**). The wound in the right trochanteric region (**e**) was closed under tension when compared with that in the left (**b**). However, eventually the edematous soft tissue subsided, and tissue tension decreased due to the dermatotraction effect of CiNPWT (**e**,**i**). Obviously, as the PC + CiNPWT flap procedure was less difficult, the wound length was significantly shorter than in the pALT flap procedure. Abbreviations: CiNPWT, closed-incision negative wound pressure therapy; PC, primary closure; VAC, vacuum-assisted closure; pALT, pedicle anterolateral thigh flap.

**Table 1 jpm-12-00182-t001:** Overall patient characteristics.

Variable	Total Patients (*n* = 128)	
	*n*	%
Age (Mean ± SD)	75.80 ± 14.71	
Sex		
Male	60	46.88
Female	68	53.13
Etiology of bedridden state		
Dementia	36	28.13
Cerebrovascular accident	30	23.44
Parkinson’s disease	14	10.94
Spinal cord injury	24	18.75
Fracture of femur	13	10.16
Other	11	8.59
**Variable**	**Total Wounds (*n*_T_ = 162)**	
	** *n* ** ** _T_ **	**%**
Location of defect		
Sacral	114	70.37
Trochanter	31	19.14
Ischial	7	4.32
Back	5	3.09
Other	5	3.09

**Table 2 jpm-12-00182-t002:** Compares of outcome in overall result.

Variable	Total Patients (*n* = 128)							
	VY (*n* = 56)	SGAP (*n* = 13)	pALT (*n* = 6)	TFL (*n* = 4)	PC (*n* = 17)	Other (*n* = 2)	PC + CiNPWT (*n* = 30)	*p* Value
Sex ^a^								0.7220
Male	27	7	3	1	5	1	16	
Female	29	6	3	3	12	1	14	
Age (Mean ± SD) ^b^	76.93 ± 12.35	74.15 ± 17.86	61.17 ± 20.19	75.75 ± 14.41	78.29 ± 12.37	54.50 ± 33.23	77.33 ± 15.08	0.0715
Case follow-up time (month, Mean ± SD)	20.40 ± 19.24	12.96 ± 14.03	13.38 ± 14.81	33.88 ± 24.98	22.32 ± 17.64	24.00 ± 2.83	11.4 3± 8.74	0.0546
Comorbidity ^a^								
HTN	33	7	2	3	10	1	17	0.9218
CHF	5	0	0	0	3	0	1	0.5594
VHD	3	0	0	1	0	0	2	0.4590
CAD	9	0	0	0	2	0	3	0.7590
Arrhythmia	5	1	1	0	1	0	3	0.9478
Dyslipidemia	3	0	1	0	1	0	4	0.5472
PAOD	2	0	0	0	1	0	1	0.9220
DM	22	4	2	3	7	1	13	0.8478
CKD	9	1	0	0	1	0	3	>0.999
ESRD HD	3	0	0	0	0	0	1	>0.999
Other	8	1	1	0	2	0	4	>0.999
Nil	10	6	3	1	4	1	8	0.2137
**Variable**	**Total Wounds (*n*_T_ = 162)**							
	**VY (*n*_T_ = 66)**	**SGAP (*n*_T_ = 14)**	**pALT (*n*_T_ = 8)**	**TFL (*n*_T_ = 10)**	**PC (*n*_T_ = 30)**	**Other (*n*_T_ = 2)**	**PC + CiNPWT (*n*_T_ = 32)**	** *p* ** **Value**
Wound size (cm^2^, Mean ± SD) ^b^	64.24 ± 56.88	103.29 ± 36.31	92.38 ± 31.01	56.00 ± 40.52	35.03 ± 23.13	67.50 ± 60.10	63.47 ± 42.70	<0.001
Operation time (min, Mean ± SD) ^b^	78.85 ± 26.82	126.00 ± 32.26	174.88 ± 45.16	121.60 ± 61.13	38.33 ± 12.70	141.00 ± 124.45	38.16 ± 14.02	<0.001
Debridement time (Mean ± SD) ^b,c^	2.50 ± 1.83	3.79 ± 4.35	4.50 ± 2.39	2.60 ± 1.51	2.40 ± 1.25	3.50 ± 2.12	2.13 ± 0.98	0.0229
Hospital days (Mean ± SD) ^b^	60.92 ± 73.39	63.15 ± 61.55	61.00 ± 25.60	45.80 ± 27.84	47.00 ± 26.10	58.00 ± 65.05	36.78 ± 26.92	0.4906
Location of defect								
Sacral	65	14	0	0	11	0	24	
Trochanter	1	0	6	10	8	0	6	
Ischial	0	0	2	0	4	0	1	
Back	0	0	0	0	3	1	1	
Other	0	0	0	0	4	1	0	

*n* = numbers of patients; *n*_T_ = numbers of wounds; ^a^ Chi-square test or Fisher’s exact test. ^b^ One-way analysis of variance (ANOVA). ^c^ Debridement time included the debridement before reconstruction and the last debride during reconstruction. Abbreviations: CAD = coronary artery disease, CHF = congestive heart failure, CKD = chronic kidney disease, CVA = cerebral vascular accident, DM = diabetes mellitus, ESRD = end-stage renal disease, HD = hemodialysis, HTN = hypertension, PAOD = peripheral arterial occlusive disease, VHD = valvular heart disease.

**Table 3 jpm-12-00182-t003:** Compares of outcome between Traditional and PC + CiNPWT groups.

Variable	Total Patients (*n* = 128)				
	Traditional Reconstruction (*n* = 98)		PC + CiNPWT (*n* = 30)		*p* Value
	*n*	%	*n*	%	
Sex ^a^					0.4179
Male	44	44.9	16	53.33	
Female	54	55.1	14	46.67	
Age (Mean ± SD) ^b^	75.33 ± 14.65		77.33 ± 15.08		0.5155
**Variable**	**Total Wounds (*n*_T_ = 162)**				
	**Traditional Reconstruction (*n*_T_ = 130)**		**PC + CiNPWT (*n*_T_ = 32)**		***p* Value**
	** *n* _T_ **	**%**	** *n* _T_ **	**%**	
Wound size (cm^2^, Mean ± SD) ^b^	62.85 ± 49.94		63.47 ± 42.70		0.9490
Operation time (min, Mean ± SD) ^b^	84.73 ± 48.55		38.16 ± 14.02		<0.001
Debridement time (Mean ± SD) ^b^	2.76 ± 2.20		2.13 ± 0.98		0.0153
Hospital days (Mean ± SD) ^b^	56.70 ± 58.43		36.78 ± 26.92		0.0054
Outcome ^a^					
Healed	98	75.38	26	81.25	0.4830
Minor	28	21.54	6	18.75	0.7286
Major	5	3.85	0	0	0.2598
Mortality	13	10.00	2	6.25	0.7374
Location of defect ^a^					0.9641
Sacral	90	69.23	24	75	
Trochanter	25	19.23	6	18.75	
Ischial	6	4.62	1	3.13	
Back	4	3.08	1	3.13	
Other	5	3.85	0	0	

^a^ Chi-square test or Fisher’s exact test. ^b^ Independent *t* test.

**Table 4 jpm-12-00182-t004:** Compares of outcome between PC and PC + CiNPWT groups.

Variable	Total Wounds (*n*_T_ = 62)				
	PC (*n*_T_ = 30)		PC + CiNPWT (*n*_T_ = 32)		*p* Value
	*n* _T_	%	*n* _T_	%	
Wound size (cm^2^, Mean ± SD) ^a^	35.03 ± 23.13		63.47 ± 42.70		0.0019
Operation time (min, Mean ± SD) ^a^	38.33 ± 12.70		38.16 ± 14.02		0.9587
Debridement time (Mean ± SD) ^a^	2.40 ± 1.25		2.13 ± 0.98		0.3360
Hospital days (Mean ± SD) ^a^	47.00 ± 26.10		36.78 ± 26.92		0.1349
Outcome ^b^					
Healed	28	93.33	26	81.25	0.2577
Minor	4	13.33	6	18.75	0.7331
Major	1	3.33	0	0	0.4839
Mortality	1	3.33	2	6.25	0.5928
Location of defect ^b^					0.0109
Sacral	11	36.67	24	75	
Trochanter	8	26.67	6	18.75	
Ischial	4	13.33	1	3.13	
Back	3	10	1	3.13	
Other	4	13.33	0	0	

^a^ Independent *t* test. ^b^ Chi-square test or Fisher’s exact test.

## Data Availability

The data presented in this study are available in Appendix A.

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
