# Peer review of "Simple and Efficient Pressure Ulcer Reconstruction via Primary Closure Combined with Closed-Incision Negative Pressure Wound Therapy (CiNPWT)—Experience of a Single Surgeon"

_jpm, 2022, doi:10.3390/jpm12020182_

Round 1

Reviewer 1 Report

The authors are to be commended for submitting results in pressure ulcer reconstruction that defy the status quo. The results presented are compelling given the co-morbid illnesses and advanced age of the patients described.  However, there are a number of issues that require addressing:

Terminology – please be consistent and refer to these wounds as pressure ulcers (bedsores is not appropriate)

Introduction:

Note that pressure ulcers also come from sitting and sacral ulcers in particular are often related to sitting in a reclined position, not just from lying down

Rather than paraplegia, recommend using Spinal Cord Injury as that will also include quadriplegia

Please consider re-wording this sentence as it difficult to understand as currently worded: Previous studies have shown that NPWT could decrease tension across the suture site which distributes the concentrated traction forces at the suture point to the skin flap and increases skin flap perfusion.

In general the English language should undergo review as there are a number of small problems throughout.

The authors describe that a single surgeon began using ciNPWT October 2019

The retrospective series runs from January 2011 to March 2021, therefore many of the patients included were treated prior to beginning use of ciNPWT.  Given the long duration of the study it is very possible that the surgeon adapted his/her technique to achieve better outcomes. It would be a less biased study to only include October 2019-March 2021.  Furthermore, the authors have published a retrospective report of 117 Stage III/IV pressure ulcers treated between 2010 and March 2019, comparing flap closure to primary closure. [Chen CY, Chiang IH, Ou KL, Chiu YL, Liu HH, Chang CK, Wu CJ, Chu TS, Hsu KF, Huang DW, Tzeng YS. Surgical treatment and strategy in patients with pressure sores: a single-surgeon experience. Medicine 2020;99:44(e23022).]  This series is not mentioned in the current manuscript and it is unclear whether there is overlap in the patient population.  The conclusions of the 2020 publication are at odds with the current manuscript as it is stated that only smaller defects (<16cm2 are suitable for primary closure).  The difference may the institution of ciNPWT, however, this is not discussed and the 2 cohorts (PC vs PC+ciNPWT) are not compared.

There is no mention of presence or treatment of osteomyelitis and whether this was a factor in deciding to use flap reconstruction vs primary closure.

One month followup is not sufficient to evaluate lack of recurrence.

The patient population is skewed toward patients with sacral pressure ulcers.  This may be a local factor related to the patients seen at this particular hospital and not generalizable

In addition the duration of hospitalization is quite prolonged and not likely to be generalizable.

These factors must be discussed.

Author Response

Response letter to reviewer:

Reviewer 1:

Point 1

Terminology – please be consistent and refer to these wounds as pressure ulcers (bedsores is not appropriate).

Response 1

All “bedsores” were rephrased to “pressure ulcer.”

Point 2

Introduction: Note that pressure ulcers also come from sitting and sacral ulcers in particular are often related to sitting in a reclined position, not just from lying down. Rather than paraplegia, recommend using Spinal Cord Injury as that will also include quadriplegia.

Response 2

Thanks to the reviewer’s suggestions, this section has been slightly revised to make sentences more fluent and for accuracy.

Point 3

Please consider re-wording this sentence as it difficult to understand as currently worded: Previous studies have shown that NPWT could decrease tension across the suture site which distributes the concentrated traction forces at the suture point to the skin flap and increases skin flap perfusion.

Response 3

We had rewritten the sentence. It means that when a wound was sutured under tension, the soft tissue was most damaged at suture point, where the NPWT could decrease the tension by distributes the concentrated traction force to skin flap.

Point 4

In general, the English language should undergo review as there are a number of small problems throughout.

Response 4

The entire manuscript has been re-edited by a language editing company. We apologize for the unclear meaning. The revised manuscript has been re-checked and edited, and we hope that it meets your requirements. The editing certification has been attached with the e-mail.

Point 5

The authors describe that a single surgeon began using ciNPWT October 2019.

The retrospective series runs from January 2011 to March 2021, therefore many of the patients included were treated prior to beginning use of ciNPWT. Given the long duration of the study it is very possible that the surgeon adapted his/her technique to achieve better outcomes. It would be a less biased study to only include October 2019-March 2021.

Response 5

With time, surgeons’ skills also improve, which may influence statistical result. But if the retrospective time was shortened to 2019-2021, only 1 SGAP and 2 ALT cases would be obtained. With the 5-year interval of 2016-2021, we had only 8 SGAP and 3 pALT cases. Because these surgical cases are relatively small, we selected 2011 as the start of our retrospective time, when our hospital began to implement electronic operation reports, allowing us to obtain photos of surgery.

In addition, compared with other surgical procedures, the surgical skills required for pressure ulcer reconstruction surgery are not so great. Most well-trained plastic surgeons can easily complete this operation. This may not have had much impact on statistics. Regarding the length of the statistical time and possible bias on statistical results of improvement of surgical skills, we have provided an explanation in the discussion section.

Point 6

Furthermore, the authors have published a retrospective report of 117 Stage III/IV pressure ulcers treated between 2010 and March 2019, comparing flap closure to primary closure. [Chen CY, Chiang IH, Ou KL, Chiu YL, Liu HH, Chang CK, Wu CJ, Chu TS, Hsu KF, Huang DW, Tzeng YS. Surgical treatment and strategy in patients with pressure sores: a single-surgeon experience. Medicine 2020;99:44(e23022).] This series is not mentioned in the current manuscript and it is unclear whether there is overlap in the patient population. The conclusions of the 2020 publication are at odds with the current manuscript as it is stated that only smaller defects (<16cm2 are suitable for primary closure). The difference may the institution of ciNPWT, however, this is not discussed and the 2 cohorts (PC vs PC+ciNPWT) are not compared.

Response 6

With respect to a previous study on similar population, I have added a paragraph in the discussion section. I have compared PC and PC+CiNPWT cohorts in the manuscript (Section: Comparison of outcomes between the PC and PC+CiNPWT groups. Line 261-271) and have also provided a table (Table 4).

Point 7

There is no mention of presence or treatment of osteomyelitis and whether this was a factor in deciding to use flap reconstruction vs primary closure.

Response 7

For osteomyelitis of pressure ulcer, most cases are caused by bone exposure and wound infection. In our experience, osteomyelitis mostly exists in the surface cortical bone, whereas more severe case may involve invasion to the medullary bone. However, during debridement operations before reconstruction, we attempted to remove these sequestrum, making wound bed viable. With appropriate systemic antibiotic treatments, most showed good results. In our pressure ulcer routine treatment, when there was wound infection, we administered antibiotic treatment, along with several debridement operations (about 2-3 times, there may be more in difficult cases), and continued to administer after wound reconstruction until suture removal. Generally, patients with this protocol received antibiotic treatment for more than 4–6 weeks (the interval between each debridement is about 1 week, plus about 2–3 weeks after reconstruction till suture removal).

Our treatment method was also consistent with the literature [Wong D, Holtom P, Spellberg B. Osteomyelitis Complicating Sacral Pressure Ulcers: Whether or Not to Treat With Antibiotic Therapy. Clin Infect Dis. 2019 Jan 7;68(2):338-342. doi: 10.1093/cid/ciy559. PMID: 29986022; PMCID: PMC6594415.] and [Andrianasolo J, Ferry T, Boucher F, Chateau J, Shipkov H, Daoud F, Braun E, Triffault-Fillit C, Perpoint T, Laurent F, Mojallal AA, Chidiac C, Valour F; Lyon BJI study group. Pressure ulcer-related pelvic osteomyelitis: evaluation of a two-stage surgical strategy (debridement, negative pressure therapy and flap coverage) with prolonged antimicrobial therapy. BMC Infect Dis. 2018 Apr 10;18(1):166. doi: 10.1186/s12879-018-3076-y. PMID: 29636030; PMCID: PMC5894174.].

The presence of combined osteomyelitis in the wound was not a factor in whether we chose PC+CiNPWT to reconstruct the wound. On the other hand, when severe osteomyelitis was present, most wounds were large with large area of bone exposure. It may not be possible to reconstruct the wound simply with PC+CNPWT. We chose PC+CiNPWT because "this way the wound can be closed, and it can shorten the operation time and reduce the risk of anesthesia". This was our primary goal.

Point 8

One-month follow-up is not sufficient to evaluate lack of recurrence.

Response 8

All cases discharged from our department had an OPD visit in the following months for wound follow up. The follow-ip times for most cases were more than 3 months, except some cases that did not follow the appointment. On the other hand, for some problematic cases such as patients with low social-economic level or older patients living alone with poor selfcare ability, we had a home-care team that provided regular home visits and medical follow-up. The average follow-up time is shown in Table 2.

The recurrence rate is an issue of pressure ulcer reconstruction, and myocutaneous flap is more resistant to pressure ulcer recurrence, which was why primary closure was rarely mentioned in traditional pressure reconstruction. The recurrence time for most reconstructed pressure ulcers ranged from few months to years and had been described in many studies. In our study, only 6 wounds of PC+CiNPWT were treated for >1 year so far. It is difficult to make a definite conclusion of recurrence rate of PC+CiNPWT. This will be the topic of our future studies.

Point 9

The patient population is skewed toward patients with sacral pressure ulcers. This may be a local factor related to the patients seen at this particular hospital and not generalizable

Response 9

With respect to the patient population skewed toward patients with sacral pressure ulcers, the incidence of bedsores in the sacral part is high (70% vs 28% [Pieper, B.A. (2010). Pressure Ulcers: Impact , Etiology , and Classifi cation]). However, this was the data of a single surgeon and did not reflect the situation of the whole department or hospital. Also, this study only collected cased who required surgery, this may indirectly indicate that pressure ulcers requiring surgical reconstruction are mostly concentrated in the sacral region. This may be followed by more detailed data collection and research.

Point 10

In addition the duration of hospitalization is quite prolonged and not likely to be generalizable. These factors must be discussed.

Response 10

Duration of hospitalization has always been the most difficult problem in the service of physicians in public hospitals. Due to the national health insurance in our contry, patients only need to bear a very low hospitalization cost. In contrast, if patients are discharged from the hospital for home care or referred to other care institutions, they have to bear a huge cost. For example, for patients with pressure ulcers, most family members will request discharge after the wound has totally healed, and only a few can accept suture removal in the outpatient clinic. Seldom, patients and families can accept being discharged from the hospital a few days after reconstruction and maintain OPD follow-up. Therefore, patients who require long-term care are often stranded in the hospital. This is a unique medical situation in the country and needs to be reformed.

Reviewer 2 Report

A manuscript: „Simplify and Speed Up Pressure Ulcer Reconstruction with Simple Primary Closure Combined Closed Incision Negative Pressure Wound Therapy (CiNPWT)” is an innovative and interesting view into a problem of decubitus patients who are not suitable for longer general anesthesia. The authors aimed to report the use of CiNPWT combined with the primary ulcer closure to analyze the clinical efficacy of this procedure as an alternative to flap surgery. The methodology is retrospective but the number of cases sufficient to show promising results that are worth further investigation with improved methods. I have several remarks that are listed below:

Line 59: Please explain in few words the difference between NPWT and CiNPWT.

Lines 73-77: Authors state that when no residual necrotic tissue after wound debridement was present, the patient was eligible for CiNPWT. However later in other parts of the manuscript it is stated and shown in photographs that the wounds were excised surgically before direct closure. This surgical part of treatment should be described in Methods in more details. If wound excision was performed simultaneously with wound closure and application of CiNPWT when these decisions were made: before or during surgery? How the wound edges were approximated?

Lines 82-83: What if wound fluid discharge was active for longer than 14 days? How many days after CiNPWT unit disassembling the patients were discharged? Have you been using CiNPWT or regular vacuum drains after traditional flap reconstruction? When the decisions concerning patient hospital discharge were made in each group? This is unclear in your manuscript.

Lines 95-98. Authors state: “… 29 patients were excluded due to incomplete records, including photographs of wound reconstruction, medical records of the wound healing process during admission or at the outpatient department, and photos of wound complications if present”. Are these the exclusion criteria? This sentence is incomprehensible.

Lines 100-101: Authors state that VAC application on reconstructed wounds, hospital stay, and wound complications, were recorded. Please define “VAC application time” (was it CiNPWT, or regular NPWT or regular vacuum drains?) and consider adding this information to your Results.

Table 2: Explain the acronyms in the comorbidity column in Table 2. Also state what units were used for wound size and time in Table 2.

Table 2: How do you define debridement time? Is this a total time of local debridements performed before operation? Or maybe this is wound excision time required to excise the wound during the operation? The difference between local bedside debridement and operative wound excision is completely unclear in your manuscript. Please define both procedures and change methods accordingly.

Case reports: The authors state that cases that were described (cases 1-3) were observed 1-2 months after the reconstruction. Consider adding to the table 2 what was the mean follow up after operations ?

Case 4: The patient had left trochanteric region reconstructed with a pedicle ALT flap and the sacral and right trochanteric regions closed primarily with CiNPWT. To which group in Table 2 this patient was included? How many patients treated simultaneously with a flap and CiNPWT were included? How such patients were compared with other patients treated with one method only? Please explain this in your manuscript or change Methods accordingly.

Lines 235-237: When authors sum up all wounds reconstructed with different methods there is no difference in average reconstructed wound size between other reconstruction methods and PC+CiNPWT. It probably results from relatively small wound sizes in PC group, TFL group, VY group and Other group. The groups of SGAp and pALT had largest reconstructed wound sizes, longest operative reconstruction time and the largest number of debridements. If SGAp and pALT groups alone would be summed up and compared with PC+CiNPWT the results might be different. Perhaps therefore PC+CiNPWT is suitable only for certain, well chosen wounds with no indications for larger flaps. Please consider adding few remarks concerning this problem in Discussion.

Lines 264-265: What type of anaesthesia was used during wound excision for PC+CiNPWT application. Authors should mention this in Methods.

Lines 295-298: This is an interesting but difficult problem. Secondary wound dehiscence after the first wound excision and reconstruction often requires prolonged treatment with NPW, only to clean the remaining ulcer. Authors state here that: most dehiscence wounds can be successfully reconstructed through the second PC+CiNPWT procedure. This problem however was not mentioned in current manuscript Results. Is this really possible to primarily close such wounds in all cases? I suggest adding few words of explanation here or even reserving discussion of this problem for future research and next publications.

Primary closure of decubitus ulcer is not routinely used for several reasons, including postoperative scar that remains directly over bony prominence. Long-term results of such a simplified treatment may be worse than flap reconstruction. Please consider mentioning this problem in Discission.  

Author Response

Response letter to reviewer:

Reviewer 2:

Point 1

Line 59: Please explain in few words the difference between NPWT and CiNPWT.

Response 1

Negative pressure wound therapy is applied on open wound for temporal wound coverage and drainage. It is used for bridge unstable wounds for reconstruction. Close incision NPWT is a extended use of NPWT: the NPWP’s traction effect approximates both skin flaps and increases tissue perfusion, therefore improving wound healing.

Point 2

Lines 73-77: Authors state that when no residual necrotic tissue after wound debridement was present, the patient was eligible for CiNPWT. However later in other parts of the manuscript it is stated and shown in photographs that the wounds were excised surgically before direct closure. This surgical part of treatment should be described in Methods in more details. If wound excision was performed simultaneously with wound closure and application of CiNPWT when these decisions were made: before or during surgery? How the wound edges were approximated?

Response 2

I have revised this paragraph and added relevant details about debridement, antibiotic treatment, reconstruction method decision, wound suture method, and wound follow-up. Thanks for the advice.

Point 3

Lines 82-83: What if wound fluid discharge was active for longer than 14 days? How many days after CiNPWT unit disassembling the patients were discharged? Have you been using CiNPWT or regular vacuum drains after traditional flap reconstruction? When the decisions concerning patient hospital discharge were made in each group? This is unclear in your manuscript.

Response 3

If wound fluid discharge was present for longer than 14 days, there may be three reasons: First, if drainage was fresh or dark red, it may indicate active bleeding or soft tissue oozing; second, if drainage was clear, it can be seroma, which will cause wound poor healing or dehiscence later; third, if the drainage was turbid, with an odor, this may indicate an infection. All three conditions need further surgery, then this case will be included in the complication.

Point 4

Lines 95-98. Authors state: “… 29 patients were excluded due to incomplete records, including photographs of wound reconstruction, medical records of the wound healing process during admission or at the outpatient department, and photos of wound complications if present”. Are these the exclusion criteria? This sentence is incomprehensible.

Response 4

I have rewritten this paragraph.

Point 5

Lines 100-101: Authors state that VAC application on reconstructed wounds, hospital stay, and wound complications, were recorded. Please define “VAC application time” (was it CiNPWT, or regular NPWT or regular vacuum drains?) and consider adding this information to your Results.

Response 5

In line 100-101, “VAC application on reconstructed wounds”: VAC means CiNPWT. I have revised the term VAC to CiNPWT in the manuscript, and added more discussion on VAC usage. We use VAC as temporalized wound coverage between debridement; however, we did not collect data on the length of VAC applied on the wound, and this was not the goal of this study. This is a topic worth discussing, and maybe more research can be done in the future.

Point 6

Table 2: Explain the acronyms in the comorbidity column in Table 2. Also state what units were used for wound size and time in Table 2.

Response 6

Explanation of acronyms have been added below the table, and unit of wound size and time were added.

Point 7

Table 2: How do you define debridement time? Is this a total time of local debridement performed before operation? Or maybe this is wound excision time required to excise the wound during the operation? The difference between local bedside debridement and operative wound excision is completely unclear in your manuscript. Please define both procedures and change methods accordingly.

Response 7

About the debridement time and operation procedure, debridement time is the total debridement before reconstruction, including the last debridement during reconstruction. I have made revisions on the surgical method and have explained debridement time under Table 2. All the debridement in the paragraph means surgical debridement under anesthesia.

Point 8

Case reports: The authors state that cases that were described (cases 1-3) were observed 1-2 months after the reconstruction. Consider adding to the table 2 what was the mean follow up after operations ?

Response 8

I have added the follow-up time in Table 2.

Point 9

Case 4: The patient had left trochanteric region reconstructed with a pedicle ALT flap and the sacral and right trochanteric regions closed primarily with CiNPWT. To which group in Table 2 this patient was included? How many patients treated simultaneously with a flap and CiNPWT were included? How such patients were compared with other patients treated with one method only? Please explain this in your manuscript or change Methods accordingly.

Response 9

I have added the grouping details in the paragraph: "Result-Comparison of overall outcomes".

Point 10

Lines 235-237: When authors sum up all wounds reconstructed with different methods there is no difference in average reconstructed wound size between other reconstruction methods and PC+CiNPWT. It probably results from relatively small wound sizes in PC group, TFL group, VY group and Other group. The groups of SGAp and pALT had largest reconstructed wound sizes, longest operative reconstruction time and the largest number of debridement. If SGAp and pALT groups alone would be summed up and compared with PC+CiNPWT the results might be different. Perhaps therefore PC+CiNPWT is suitable only for certain, well chosen wounds with no indications for larger flaps. Please consider adding few remarks concerning this problem in Discussion.

Response 10

I agree with you, this statistical omission was due to our negligence, and we have added remarks in the discussion.

Point 11

Lines 264-265: What type of anaesthesia was used during wound excision for PC+CiNPWT application. Authors should mention this in Methods.

Response 11

The type of anesthesia used was determined by the specialist anesthesiologist, based on the patient's physical condition, surgical site, posture, and surgery. General or spinal anesthesia both meet our surgical needs. If the patient cooperated, the operation also could completed under light sedation with tumescent anesthesia. These sentences have been added to the Method section, line 93-96.

Point 12

Lines 295-298: This is an interesting but difficult problem. Secondary wound dehiscence after the first wound excision and reconstruction often requires prolonged treatment with NPW, only to clean the remaining ulcer. Authors state here that: most dehiscence wounds can be successfully reconstructed through the second PC+CiNPWT procedure. This problem however was not mentioned in current manuscript Results. Is this really possible to primarily close such wounds in all cases? I suggest adding few words of explanation here or even reserving discussion of this problem for future research and next publications.

Response 12

About the sentence “Even if subsequent wounds exhibit partial dehiscence, the edges of tight wounds would also relax due to the dermatotraction effect of NPWT; accordingly, most dehiscence wounds can be successfully reconstructed through the second PC+CiNPWT procedure”. It means while partial dehiscence wounds were not too big, and there was no serious infection or considerable unviable tissue, it is feasible to directly excise the unhealthy tissue at the wound edge, and perform PC+CiNPWT. Of course, if the dehiscence wound had too much dead space, necrotic tissue, odor smelling, or wound infection, the standard wound reconstruction steps: including debridement, NPWT and follow-up reconstruction surgery are still needed, and these cases will referred as major complications.

Point 13

Primary closure of decubitus ulcer is not routinely used for several reasons, including postoperative scar that remains directly over bony prominence. Long-term results of such a simplified treatment may be worse than flap reconstruction. Please consider mentioning this problem in Discission.

Response 13

I agree with you, and we are collecting the related follow-up result. In our study, only 6 PC+CiNPWT wounds were treated for more than 1 year so far. It is difficult to make a definite conclusion of the recurrence rate and lon-term complication of PC+CiNPWT. We are preparing to study the topic in following years.

Round 2

Reviewer 1 Report

English grammar – the manuscript still contains multiple syntax errors including inappropriate and inconsistent past/present tense, plurals vs singular nouns

The new section on pages 2-3 must be reviewed for English syntax

Terminology of pressure ulcer, pressure sores and ‘bedsores’ is still inconsistent

P10/16 states that the PC+ciNPWT underwent the fewest number of debridements. Please clarify whether this indicates lesser severity in this group of wounds.

If consistent with the intent, please change ‘good wound healing’ to ‘healed’ in the text and in Tables 3 and 4 under ‘Outcome’

Recommend changing ‘long-term’ anesthesia to lengthy or prolonged anesthesia as ‘long-term’ suggests multiple days or weeks

Because the manuscript includes a lot of speculation regarding outcomes, choice of procedure and non-standard wound bed preparation (betadine wet dressings) it should be emphasized that this is a single surgeon’s experience, including altering the title to reflect that emphasis.

Example: A single surgeon’s experience with simple primary closure combined with closed incision negative pressure wound therapy

Also the discussion should explain that primary closure with NPWT was performed in selected cases

The discussion of osteomyelitis in the response to review should be included in the manuscript. Opinions on treatment and extent of osteomyelitis in pressure ulcers remains quite variable.

I would recommend statistical review and review by an anesthesiologist.  Most anesthesiologists that I know would be reluctant to perform spinal anesthesia in a patient with a large sacral wound and spinal anesthesia in older adults can be very difficult. A recent publication from orthopedic surgery/hip replacements suggest no benefit in terms of post-op outcomes for spinal vs general anesthetic

Author Response

Dear reviewer:

In accordance with the comments provided, we have substantially revised the text for both language and content. We have provided careful explanations regarding the limitations of the study, and we have ensured consistent and accurate use of terminology in the revised version. The manuscript has also been submitted for professional editing by a native speaker of English with expertise in the subject matter. Our point-by-point responses to the reviewers’ concerns are included along with a copy of the revised manuscript. We believe that these revisions have substantially improved our paper, which we hope is now suitable for publication in your esteemed journal.

Response to reviewer point-by-point:

Point 1

English grammar – the manuscript still contains multiple syntax errors including inappropriate and inconsistent past/present tense, plurals vs singular nouns. The new section on pages 2-3 must be reviewed for English syntax

Response 1

We thank the reviewer for this honest critique of our manuscript. The entire manuscript has been re-checked by a native speaker of English with expertise in the subject matter.

Point 2

Terminology of pressure ulcer, pressure sores and ‘bedsores’ is still inconsistent

Response 2

We apologize for these initial inconsistencies. All instances of “pressure sore” and “bed sore” have been revised to “pressure ulcer”.

Point 3

P10/16 states that the PC+ciNPWT underwent the fewest number of debridements. Please clarify whether this indicates lesser severity in this group of wounds.

Response 3

We thank the reviewer for highlighting this important issue. Notably, it is possible that differences in debridement times and hospitalization length may have been due to differences in general condition and comorbidities among patients. Patients whose wounds can be reconstructed with PC may be in relatively better condition, with less severe wounds and fewer comorbidities. However, our statistical analyses revealed no differences in sex, age, or comorbidities between groups. Therefore, considering the objective data, general condition, wound condition, and comorbidities are not the main factors affecting debridement times and hospitalization length. In a typical clinical setting, NPWT is used for temporalized wound coverage between debridements when affordable for the patient. Many studies have demonstrated that NPWT can shorten wound healing time,26 indicating that an associated decrease in the length of hospitalization may have biased our results. However, we did not collect data regarding the use of NPWT or its duration, necessitating a focus on these variables in future studies. Accordingly, this information has been included in the revised manuscript (lines 379-390).

Point 4

If consistent with the intent, please change ‘good wound healing’ to ‘healed’ in the text and in Tables 3 and 4 under ‘Outcome’

Response 4

We thank the reviewer for this helpful suggestion. The text has been revised as follows: “To evaluate clinical outcomes, we defined “healed” as adequate wound healing with suture removal 14–21 days after closure and no occurrence of dehiscence within the following 3 months.” We have also revised the text in Tables 3 and 4 and when discussing the comparison of outcomes between the traditional and PC+CiNPWT groups.

Point 5

Recommend changing ‘long-term’ anesthesia to lengthy or prolonged anesthesia as ‘long-term’ suggests multiple days or weeks

Response 5

We thank the reviewer for this helpful suggestion regarding terminology. Accordingly, we have revised “long-term anesthesia” to “prolonged anesthesia”.

Point 6

Because the manuscript includes a lot of speculation regarding outcomes, choice of procedure and non-standard wound bed preparation (betadine wet dressings) it should be emphasized that this is a single surgeon’s experience, including altering the title to reflect that emphasis. Example: A single surgeon’s experience with simple primary closure combined with closed incision negative pressure wound therapy

Response 6

We thank the reviewer for this important reminder. We have revised the title to indicate that the study focused on a single surgeon’s experience. This is also mentioned as a limitation of the study in the Discussion section.

Point 7

Also the discussion should explain that primary closure with NPWT was performed in selected cases

Response 7

In accordance with your recommendation, we have emphasized this point in the limitations section of the Discussion.

Point 8

The discussion of osteomyelitis in the response to review should be included in the manuscript. Opinions on treatment and extent of osteomyelitis in pressure ulcers remains quite variable.

Response 8

We thank the reviewer for this pertinent suggestion. The relevant paragraph has been incorporated into the Discussion. Thanks for suggestion. We had add the paragraph into discussion (Line: 391-413).

Point 9

I would recommend statistical review and review by an anesthesiologist. Most anesthesiologists that I know would be reluctant to perform spinal anesthesia in a patient with a large sacral wound and spinal anesthesia in older adults can be very difficult. A recent publication from orthopedic surgery/hip replacements suggest no benefit in terms of post-op outcomes for spinal vs general anesthetic

Response 9

We thank the reviewer for this pertinent suggestion. Indeed, some evidence has indicated that there are no significant differences in postoperative outcomes between patients receiving spinal anesthesia and general anesthesia during hip surgery. We also agree that the administration of spinal anesthesia can be difficult in some patients with pressure ulcers. At our center, each operation performed under anesthesia must be fully discussed with the anesthesiologist before the procedure. Our surgeons fully respect the professional judgment made by the anesthesiologist based on the patient's condition and have achieved good results according to clinical outcomes. In this article, we only present the advantages of this surgical approach, especially as an alternative for reconstruction when patients are not suitable for extended anesthesia. However, we agree that further statistical evidence is required.

Reviewer 2 Report

After author's revision Methods and Results are improved and the manuscript is understandable and very interesting.

Author Response

Thank you for your affirmation.